# Simplified Cell Magnetic Isolation Assisted SC^2^ Chip to Realize “Sample in and Chemotaxis Out”: Validated by Healthy and T2DM Patients’ Neutrophils

**DOI:** 10.3390/mi13111820

**Published:** 2022-10-25

**Authors:** Xiao Yang, Chaoru Gao, Yong Liu, Ling Zhu, Ke Yang

**Affiliations:** 1Anhui Institute of Optics and Fine Mechanics, Hefei Institutes of Physical Science, Chinese Academy of Sciences, Hefei 230031, China; 2School of Biomedical Engineering, Anhui Medical University, Hefei 230032, China

**Keywords:** neutrophil, microfluidic chip, isolation, chemotaxis test, diabetes mellitus type 2 (T2DM)

## Abstract

Neutrophil migration in tissues critically regulates the human immune response and can either play a protective role in host defense or cause health problems. Microfluidic chips are increasingly applied to study neutrophil migration, attributing to their advantages of low reagent consumption, stable chemical gradients, visualized cell chemotaxis monitoring, and quantification. Most chemotaxis chips suffered from low throughput and fussy cell separation operations. We here reported a novel and simple “sample in and chemotaxis out” method for rapid neutrophils isolation from a small amount of whole blood based on a simplified magnetic method, followed by a chemotaxis assay on a microfluidic chip (SC^2^ chip) consisting of six cell migration units and six-cell arrangement areas. The advantages of the “sample in and chemotaxis out” method included: less reagent consumption (10 μL of blood + 1 μL of magnetic beads + 1 μL of lysis buffer); less time (5 min of cell isolation + 15 min of chemotaxis testing); no ultracentrifugation; more convenient; higher efficiency; high throughput. We have successfully validated the approach by measuring neutrophil chemotaxis to frequently-used chemoattractant (i.e., fMLP). The effects of D-glucose and mannitol on neutrophil chemotaxis were also analyzed. In addition, we demonstrated the effectiveness of this approach for testing clinical samples from diabetes mellitus type 2 (T2DM) patients. We found neutrophils’ migration speed was higher in the “well-control” T2DM than in the “poor-control” group. Pearson coefficient analysis further showed that the migration speed of T2DM was negatively correlated with physiological indicators, such as HbA1c (−0.44), triglyceride (−0.36), C-reactive protein (−0.28), and total cholesterol (−0.28). We are very confident that the developed “sample in and chemotaxis out” method was hoped to be an attractive model for analyzing the chemotaxis of healthy and disease-associated neutrophils.

## 1. Introduction

Neutrophils are the first line of defense against microbial invasion and are very important in resisting diseases and protecting the host [1,2]. Dysregulation of neutrophil chemotactic signaling could cause function disorders, leading to autoimmune diseases and even fatal hazards. Researchers have previously identified neutrophil dysfunction in diabetes mellitus type 2 (T2DM) patients, such as impaired migration and chemotaxis [3,4,5]. In the last ten years, there has been a longstanding interest to develop assays for neutrophil chemotaxis. Microfluidic chips have been extensively used for investigating neutrophil chemotaxis owing to the advantages in real-time visualization, precise generation of chemokine gradient, and low consumption of samples and reagents. Many microfluidic chips have been designed to study neutrophil chemotaxis in immunocompromised patients [6,7,8,9,10,11,12]. For example, Daniel Irimia et al. developed a microfluidic chip to measure the spontaneous neutrophil motility of sepsis patients from a drop of blood [6,13]. Francis Lin et al. developed a microfluidic chip filled with a three-dimensional gel matrix to study the T cells’ transendothelial migration response to physiological skin inflammatory substances and anti-inflammatory therapeutic drugs [14,15,16]. David J. Beebe et al. invented a microfluidic chip to study how the paracrine signaling of the immune cells guided the neutrophils to migrate toward the A. fumigatus [17]. We previously integrated a microfluidic chip with a smartphone to study neutrophil migration in patients with chronic obstructive pulmonary disease (COPD) [18,19,20]. These researches laid a microfluidic-based theoretical and technical foundation for studying disease-related neutrophil chemotaxis.

Recently, some researchers have reported high throughput microfluidic chips for improving the efficiency of the chemotaxis assay [21,22,23,24]. For example, Berthier et al. developed a microfluidic chip consisting of 50 independent gradient generation units to study neutrophil chemotaxis in patients with severe recurrent bacterial infections [21]. Wu et al. described a microfluidic chip with eight independent units allowing for multiple chemotaxis assays [25,26]. Satti et al. further improved the throughput by integrating twelve channels in the microfluidic chip, which used centrifugal force to align the initial position of neutrophils [27]. Usually, microfluidic chip-based migration and chemotaxis analysis required a real-time microscope to record and track the cell migration paths [13,24,28]. Due to the limited FOV of the microscope, most “high-throughput” microfluidic chips only showed an independent channel under a FOV. Although enlarging the number of microscopes or rotating/moving the objective can improve throughput, these approaches will cause device-to-device variations, increase the cost and prolong the time-consuming for running multiple devices [21,22,23,25,27,29]. In addition, imaging and tracking fast-moving cells, such as the fastest neutrophils, often requires imaging every 10 s. Time consumption for moving the microscope stage and imaging limits the number of assays, which is not scalable for high throughput applications such as drug testing and drug screening. In addition, “high-throughput” microfluidic chips often did not possess the ability or equip an accessory to isolate cells, limiting clinical diagnostic applications [7,8,15,16,30]. Multiple chemotaxis assays often required neutrophils separated by density gradient centrifugation, which was laborious and time-consuming, required a lot of whole blood, and may lead to cell activation and damage [31,32]. Commercial kit (17957, STEMCELL, Vancouver, Canada) facilitated microfluidic chip-based cell migration studies, but still required centrifugation and had a higher reagent cost. The integration of red blood cell filters or cell-affinity antibodies on the microfluidic chips enabled neutrophil isolation from the blood of burn, asthma, COPD, and T2DM patients [12,15,33,34,35,36,37]. However, the manufacturing and pre-processing of microfluidic chips were complicated. Furthermore, multiple washing steps (requiring skilled operators) were necessary for removing the sizeable red blood cells, which potentially caused the loss of neutrophils.

We here developed a new-style “sample in and chemotaxis out” method by integrating a simplifying magnetic neutrophils isolation method with a microfluidic chip. The SC^2^ chip consisted of six cell migration channels and six-cell arrangement areas to perform multiple chemotaxis result readouts. Every independent unit of the SC^2^ chip was integrated with a cell arrangement area to adjust the position of the cells and provide reliable results output. The sizes of the six units were reduced as much as possible to fit into a 5X microscope. We used neutrophils isolated from healthy volunteers as a test model to validate the “sample in and chemotaxis out” method. The clinical application of the “sample in and chemotaxis out” method was validated by testing the neutrophil chemotaxis of T2DM patients. We observed the profoundly inhibited chemotaxis of T2DM patients’ neutrophils, consistent with previous research results [38,39]. Further, we found the migration speed of neutrophils was higher in the “well-control” (T2DM patients with HbA1c < 8) T2DM patients than in the “poor-control” (T2DM patients with HbA1c > 8). In short, our results demonstrated the potential of the “sample in and chemotaxis out” method for cell functional assays.

## 2. Materials and Methods

### 2.1. Instruments and Reagents

Neutrophil isolation kit (17957, STEMCELL, Vancouver, Canada); Bovine serum albumin (A1933-25G, Sigma-Aldrich, St. Louis, MO, USA), FBS (iCell-0500, iCell Bioscience Inc, Shanghai, China); RPMI 1640 (SH30809.01, Hyclone, Logan, UT, USA); Fluorescein (FITC-dextran, 10 kDa; final concentration 5 μM, Sigma-Aldrich, St. Louis, MO, USA); fMLP (F3506, 1 mM, Sigma-Aldrich, St. Louis, MO, USA); Triton X-100 (9036-19-5, Sigma, St. Louis, MO, MO, USA); Giemsa (G5637, Sigma, St. Louis, MO, USA); D-glucose (G7021, Sigma-Aldrich, St. Louis, MO, USA); mannitol (10 mM, an osmotic control reagent, M1902, Sigma-Aldrich, St. Louis, MO, USA); Fibronectin (F2006, 2 μg mL^−1^, Sigma-Aldrich, St. Louis, MO, USA); Commercial magnetic neutrophil isolation kit (17957, STEMCELL Technologies, Inc., Toronto, Canada); Precursors and initiators of Polydimethylsiloxane (PDMS) (Dow Corning, Midland, MI, USA); Glass slide (80350-0001, Jiangsu Shitai Xinchuang Scientific Instrument Co., Ltd., Nanjing, China); Microscope (DMi8, Leica, Wezler, Hesse, Germany); Plasma cleaner (PDC-002, Harrick Scientific Products, Inc., New York, NY, USA).

### 2.2. SC^2^ Chip Design and Fabrication

The SC^2^ chip was designed by using AutoCAD (Autodesk, San Rafael, CA, USA). The SC^2^ chip preparation steps were referenced in our previous work [20]. The master mold of the SC^2^ chip was manufactured at the USTC (University of Science and Technology of China). The master mold contained two layers. The first layer was designed to be 2 μm thick for generating the cell arrangement areas. The second layer was designed to be 60 μm thick for developing the main channels. After the master mold fabrication was completed, polydimethylsiloxane (PDMS) was injected and incubated at 70 °C for 2 h. The PDMS replica was then cut out. The PDMS with punched-out holes was bonded to a glass slide using the plasma cleaner. Before the experiments, the SC^2^ chip was filled with fibronectin and incubated at 37 °C for 60 min. The fibronectin was removed from the SC^2^ chip, where it was re-filled with the cell culture medium (0.4% bovine serum albumin diluted by RPMI-1640, Glucose-free) and incubated at 37 °C for another 60 min.

### 2.3. Gradient Simulation of the SC^2^ Chip

The Navier-Stokes equation was used to simulate the transient convection-diffusion progress between the chemokine solution and cell culture medium for gradient generation [31]. The simulation was executed using COMSOL (Version 5.5.0.359). Simulation parameters referred to previous literature [31].

### 2.4. Gradient Measurement of the SC^2^ Chip

For generating gradient, the FITC-dextran and cell culture medium were injected into the chemokine inlet and cell culture medium inlet respectively. The gradient profile developed in the migration channel was measured and analyzed by using the inverted fluorescence microscope.

### 2.5. Clinical Sample

This study was conducted according to the guidelines of the Declaration of Helsinki and approved by the Institutional Review Board (or Ethics Committee) of Hefei Institute of Physical Science, Chinese Academy of Sciences (protocol code Y-2018-21 and 17 March 2018). Whole blood of healthy donors and T2DM were gained at the Hefei Cancer Hospital, Chinese Academy of Sciences. Inclusion criteria of “poor-control” T2DM patients: clinically diagnosed diabetes, HbA1c ≥ 8%, BMI < 30 kg m^−2^. Inclusion criteria of “well-control” T2DM patients: clinically diagnosed diabetes, received proper medical treatment, HbA1c < 8%, BMI < 30 kg m^−2^. Inclusion criteria for healthy volunteers: No diabetes, HbA1c < 6.0%, no metabolic syndrome (MetS), normal BMI (18.5–24.9 kg m^−2^). Exclusion criteria: (1) severe infection, stress, cardiac insufficiency, renal insufficiency, liver damage, acute cardiovascular and cerebrovascular diseases, history of malignant tumor, mental disorders; (2) Type 1 diabetes, gestational diabetes, and other special types of diabetes; (3) Recent occurrence of acute complications of diabetes such as diabetic ketoacidosis or hyperosmolar coma; (4) Recent use of non-steroidal anti-inflammatory drugs, vitamin E and other drugs that affect high-sensitivity C-reactive protein (CRP); (5) Patients with recent severe infection, surgical experience or other stress conditions.

The volunteers with T2DM patients were required to fast for more than 8 h, and 4 mL of elbow venous blood was taken on an empty stomach at 7 am. 2 mL of whole blood was used for testing the serum levels of HbA1c, CRP, Total-C, high-density lipoprotein cholesterol (HDL-C), lipoprotein cholesterol (LDL-C), and triglycerides. The remaining 10 μL/2 mL of whole blood was used for chemotaxis analysis. The neutrophils from each donor were injected into three units of the SC^2^ chip and tested in triplicate.

### 2.6. Neutrophil Pretreatment

To measure the effects of drug stimulus on the chemotaxis of neutrophils, two parts of healthy blood samples were pre-treated with 10 mM and 15 mM of D-glucose for 1 h at 37 °C (5% CO_2_) prior to the experiment to analyze the effect of high glucose concentration on neutrophil migration [12]. Another two parts of whole blood samples were pre-treated with D-glucose (15 mM) and D-glucose (5 mM) + mannitol (10 mM) for 1 h at 37 °C (5% CO_2_) to compare the effect of D-glucose and mannitol on the chemotaxis of neutrophil. Here, mannitol was chosen as an osmotic control reagent for the D-glucose. 5 mM of D-glucose was supplemented with 10 mM of mannitol to yield a medium with a similar osmotic pressure to the 15 mM D glucose supplemented for an osmotic control.

### 2.7. Neutrophil Isolation, Chemotaxis Experiment, Analysis

For SC^2^ chip-based neutrophil isolation (Figure 1), the whole blood (10 μL) was firstly loaded separately into an Eppendorf tube. 1 μL of lysis buffer and 1 μL of magnetic beads from the commercial kit were loaded into a single Eppendorf tube and mixed with the whole blood. Then, 10 μL of 5% FBS (prepared in RPMI-1640) was added to a single Eppendorf tube. Repeat the above steps to prepare 6 samples. Six Eppendorf tubes were incubated at room temperature for 2 min. Then, magnets were vertically attached to the side of the Eppendorf tubes for 1 min. The magnetically conjugated impurities will be driven to the wall of these Eppendorf tubes, under the function of the magnetic force. Then, all medium in the inlets and outlets of the SC^2^ chip was removed. 10 μL of liquid supernatant in each Eppendorf tube with neutrophils was injected into a related cell inlet by using a pipette. Neutrophils will flow freely into the SC^2^ chip and be arranged beside the migration channel. After collecting enough neutrophils, the gradient (fMLP, prepared in RPMI-1640 with 0.4% BSA) was generated for subsequent chemotactic assay. The “sample in and chemotaxis out” method only consumed 10 μL of blood, 1 μL of magnet beads, and 1 μL of lysis buffer for single testing in a unit, the whole reagent volume was 50 times smaller than the commercial protocol (1000 μL of blood, 50 μL of magnet beads and 50 μL of lysis buffer), while providing enough neutrophils. The assay from “sample into the tubes and chemotaxis out in the chips” can be finished within 20 min.

For neutrophil chemotaxis testing, an inverted fluorescence microscope captured the cell migration images in the SC^2^ chip at 6 frames/min. All experiments were repeated three times. The image acquisition time was set at 15 min. Then, 90 images were imported into ImageJ. We rotated a unit to a horizontal position and cut out a region of interest using ImageJ before analysis. When the neutrophils migrated out of the cell arrangement area, the migration paths of neutrophils were manually tracked. At least 40 cells were analyzed. The average chemotaxis index (CI) and migration speed (V) were calculated according to the migration paths. CI was calculated as the ratio between gradient displacement and total migration distance. V was calculated as the ratio between total migration distance and movement time (15 min). In addition, we divided the 15 min migration cycle into 10 parts and calculated the time-dependent V in each 1.5 min interval. In addition, the Pearson correlation between the cell migration speed (V) and the physiological parameters (HbA1c, CRP, Total-C, HDL-C, LDL-C, Triglyceride) was analyzed by using the SPSS software. The correlation coefficient was defined as r. If |r| is calculated to tend to 1, it shows the deep correlation degree between the two parameters. If the |r| is calculated to tend to 0, it shows the poor correlation degree between the two parameters. The statistical significance was assessed the level of significance between two sets of data using the Student *t*-test with *p* < 0.05 (*) to be considered a significant correlation and *p* < 0.01 (**) to be considered a significant polarly correlation.

### 2.8. Cell Staining

Cell staining experiments confirmed the migration of neutrophils in migration channels. Specifically, neutrophils are fixed with 4% paraformaldehyde after migration, followed by infiltration of the tubes with Giemsa dye for 3 min, followed by washing with PBS.

## 3. Results and Discussion

### 3.1. SC^2^ Chip Design and Gradient Characterization

The SC^2^ chip was designed to reference past cases [20,25,27]. The SC^2^ chip was integrated with six units (Figure 2A,B). The SC^2^ chip was suitable for six chemotaxis assays simultaneously by loading various chemokines or neutrophils (Figure 2C). Every unit was designed with the following characteristics: (1) had the independent chemotactic reagent inlet, cell culture medium inlet, cell inlet, and waste outlet; (2) had a balance area to eliminate the pressure differences between the chemotactic reagent inlet and cell culture medium inlet due to minor differences in reagent volumes; (3) had a Christmas-tree-shape channel for generating an approximate linear concentration gradient in the migration channel; (4) had a cell arrangement area (2 μm of height) beside the migration channel to calibrate the initial location of the neutrophils (10 μm of diameter) (Figure 2D,E). Once the magnetically isolated neutrophils reached the cell alignment site from the cell inlet, they will be blocked beside the cell arrangement area. Once the gradients are generated, the neutrophils will polarize and transmigrate across the cell arrangement area, then chemotaxis along the migration channel (Figure 2F). Calibrating the initial positions of the neutrophils was beneficial to ensure analysis accuracy.

Both simulation and experimental tests verified the gradient profiles in the SC^2^ chip. It’s worth noting that the SC^2^ chip allowed pumpless chemoattractant gradient generation in the migration channel through laminar flow and convective diffusion. The results confirmed that the balance area was helpful for pressure balance (Figure 3A). The gradient distribution in the Christmas-tree-shaped channel was guaranteed to generate an ideal gradient in the migration channel (Figure 3A,B) [40,41]. We measured the fluorescent gradient profiles at the different points (900 μm apart) in the migration channel and found no significant difference (Figure 3C,D). We also compared the simulative and experimental fluorescent gradient profiles in the six independent units. The result showed that the simulated gradient profile matched the real fluorescent profiles (Figure 3E,F). In addition, the fluorescent gradient profiles could remain for 75 min, enough for chemotaxis testing (Figure 3E,F).

### 3.2. Characterization of “Sample in and Chemotaxis out” Method

To verify the “sample in and chemotaxis out” method, and to test the consistency of different units on the SC^2^ chip, neutrophil sorting operations were performed by a trained user. 1000 nM- and 100 nM of fMLP gradients were generated in units 1,2,3, and unit 4,5,6, separately (Figure 4A). The results indicated that the neutrophils under different conditions could quickly transmigrate across the cell arrangement areas and migrate in the migration channels (Appendix A). The V values in units 1,2,3 (units 4,5,6) were calculated to be 0.118, 0.121 and 0.108 (0.216, 0.221 and 0.221) (Figure 4B). The CI values in units 1,2,3 (units 4,5,6) were calculated to be 0.754, 0.777 and 0.761 (0.864, 0.873 and 0.859) (Figure 4C). When the same chemokine concentration gradient was generated in a different unit, the V and CI values in different units showed no significant difference. For user-friendliness, we compared the results from a trained user (technician 1, who experienced three training sessions) versus two untrained users (technicians 2 and 3, who experienced no training sessions). The experiment results are shown in Figure 4D–F. The average V and CI values gained by trained user 1 showed no significant differences from the untrained users 2 and 3. Whereafter, we carried out Giemsa nucleus staining (Figure 5). The cell staining showed the neutrophils’ ring and lobulate cores. We observed that more obvious cellular pseudopodia were distributed along the gradient direction in 100 nM fMLP than the 1000 nM fMLP (Figure 5). Appropriate fMLP concentration can bind to the corresponding high-affinity receptor proteins on the surface of the neutrophils, activating and causing neutrophil chemotaxis. As a result, the neutrophils can optimize gradient perception and chemotaxis more conveniently. On the contrary, a high concentration of fMLP would attach to the low-affinity receptor proteins, thereby releasing harmful biological substances such as lysosomal enzymes, free radicals, IL-6, etc., further inhibiting the migration of neutrophils. These results suggested that the “sample in and chemotaxis out” method was easy and friendly for chemotaxis assays. The advantages of the method included: less reagent volume (10 μL of blood + 1 μL of magnetic beads + 1 μL of lysis buffer); less time (5 min of cell isolation + 15 min of chemotaxis testing); no ultracentrifugation; on-site use; higher detection efficiency.

### 3.3. Direct On-Chip Testing of D-Glucose and Mannitol Pretreated Neutrophils’ Chemotaxis

T2DM is characterized by chronic hyperglycemia, which often causes low-grade inflammation and increases the risks of cardiovascular diseases. Hyperglycemia decreases neutrophilic functions and thus resistance to infections [42,43]. We validated the primary clinical application of the “sample in and chemotaxis out” method by testing the D-glucose pretreated neutrophil chemotaxis to a well-known chemoattractant, fMLP. The experiment result is shown in Figure 6A. As shown in Figure 6B,C, the average CI (or V) value of the healthy neutrophil was higher than the 10 mM and 15 mM D-glucose group. We further found the time-dependent V values at time points (1, 2, 3, 4, 5, 6, 7, 8, 9, 10) in the healthy group were higher than the 10 mM and 15 mM D-glucose group (Figure 6D). In addition, the time-dependent V values in the 10 mM D-glucose group were higher than the 15 mM D-glucose group at time points (3, 4, 5, 6, 7, 8, 9, 10) (Figure 6D).

We were very interested in investigating why high concentrations of D-glucose inhibited neutrophil chemotaxis. We have two guesses: ① high osmotic pressure due to high glucose levels; ② intracellular molecular interactions caused by the high concentration of D-glucose invasion into the neutrophils. The experiment result is shown in Figure 7A. As shown in Figure 7B,C, the average V and CI values of healthy neutrophils were the highest. The average V and CI values of the 5 mM D-glucose + 10 mM mannitol group were lower than the healthy group but higher than the 15 mM D-glucose group. The time-dependent V values in the healthy group were higher than in other groups at time points (1, 2, 3, 4, 5, 6, 7, 8, 9, 10) (Figure 7D). In addition, the time-dependent V values in the 5 mM D-glucose + 10 mM mannitol group were higher than the 15 mM D-glucose group at time points (1, 2, 3, 4, 5, 6, 7, 8) (Figure 7D). This experiment again demonstrated that a high level of glucose incubation was harmful to neutrophil migration. In addition, we found that the mannitol control group (5 mM D-glucose + 10 mM mannitol) still can keep a high migration speed and does not display depth inhibition on the chemotaxis of neutrophils. Thus, we assumed that the inhibited neutrophil chemotaxis seems to depend on the molecular interaction of D-glucose rather than on the osmotic pressure changes [44]. Neutrophils are metabolically active cells, and associated functions are highly energy-dependent [45]. Glycolysis is a preferred pathway to metabolize D-glucose and generate energy in neutrophils [46,47]. Our preliminary guess is that when the external D-glucose is within the concentration range of physiological concentration (5 mM D-glucose + 10 mM mannitol), the D-glucose molecule could normally flux across the plasma membrane, which is conducive to cell migration and chemotaxis. When neutrophils are incubated with a high concentration of glucose (15 mM D-glucose), excessive glucose molecules enter the cell through the cell membrane and interfere with the normal glycolysis process of the cell, affecting its chemotaxis. In short, these results demonstrated the feasibility of the developed “sample in and chemotaxis out” method for various chemotaxis assays by changing chemical stimuli.

### 3.4. Direct Testing of TD2M Patients’ Neutrophil Chemotaxis

TD2M is a metabolic disease characterized by hyperglycemia and often accompanied by various complications, such as chronic damage and dysfunction of multiple tissues, especially the eyes, kidneys, heart, blood vessels, and nerves [48]. Dysfunctions of neutrophils have been previously founded in TD2M patients [12,49]. We sought to validate the clinical applicability of the “sample in and chemotaxis out” method for synchronous chemotaxis testing from multiple samples. The experiment result is shown in Figure 8A. Specifically, we simultaneously monitored the chemotaxis of neutrophils from TD2M patients (n = 5) and one healthy donor (n = 1) as a control (Figure 8A). The results showed that the average V values of the TD2M patients (in addition to patient 2) were significantly weaker than the healthy donor (Figure 8B). Time-dependent chemotaxis analysis showed that the time-dependent V values of healthy neutrophils were higher than that in TD2M patients 1, 3, 4, and 5 (Figure 8C–F). When infection or inflammation occurs, neutrophils are activated and then migrate to the lesion area and clear the invading pathogen. Impaired neutrophil migration would lead to a decline in an individual’s immune function. Thus, we preliminarily speculated that the V of neutrophils could be used as a cell-based biomarker for individual immune function assessment and prognosis monitoring rapidly.

Then, we collected more TD2M patients (n = 12). To reduce the labor intensity of the experiment, we only invited two TD2M patients for each test. We detected the neutrophil chemotaxis of two TD2M patients on one SC^2^ chip (i.e., each TD2M patient shared three units). We simultaneously measured and compared their serum levels of HbA1c, CRP, Total-C, HDL-C, LDL-C, and triglycerides. As shown in Table 1, the results showed that the average CRP, Total-C, and triglyceride levels of the “poor-control” TD2M patients were higher than the “well-control” group. Simultaneously, the average V value of the “well-control” group was calculated to be higher than the “poor-control” group. These observed neutrophil functional phenotypes matched other microfluidics assays [12,38]. As shown in Figure 9, Pearson coefficient analysis showed that the migration speed of T2DM was negatively correlated with physiological indicators, such as HbA1c (−0.44 **), triglyceride (−0.36 **), C-reactive protein (−0.28 *) and total cholesterol (−0.28 **). We believe that the abnormal chemotactic migration function of neutrophils may be related to the abnormal inflammation and metabolic status of TD2M. We proposed decreasing the cholesterol, and other inflammatory factors of TD2M patients by dietary supplementation may help maintain normal neutrophil chemotaxis. In conclusion, we demonstrated the feasibility of the “sample in and chemotaxis out” method for the risk stratification and immune function assessment in TD2M patients.

## 4. Conclusions

In short, we demonstrated a novel “sample in and chemotaxis out” method for multiplexed chemotaxis assay, which was realized by integrating an SC^2^ chip consisting of six independent migration units with a simplified cell magnetic isolation technique. We successfully validated the performance of the current method using regular, in vitro drug-pretreated (e.g., low and high D-glucose, mannitol) and T2DM patients’ samples. We believed that this new cell chemotaxis assay model can provide more significant opinions on the correlation between abnormal cardiovascular risk factors and dysfunction of neutrophils. Moreover, the current research was still in the laboratory stage. In the following, we will further improve the “sample in and chemotaxis out” method, and apply it to the chemotaxis research of cancer cells, T cells, NK cells, and macrophages.

## Figures and Tables

**Figure 1 micromachines-13-01820-f001:**
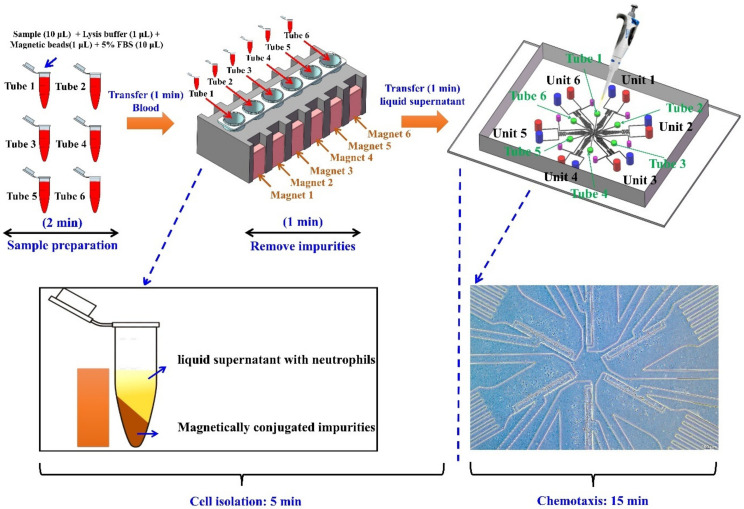
The operation progress of the “sample in and chemotaxis out” method.

**Figure 2 micromachines-13-01820-f002:**
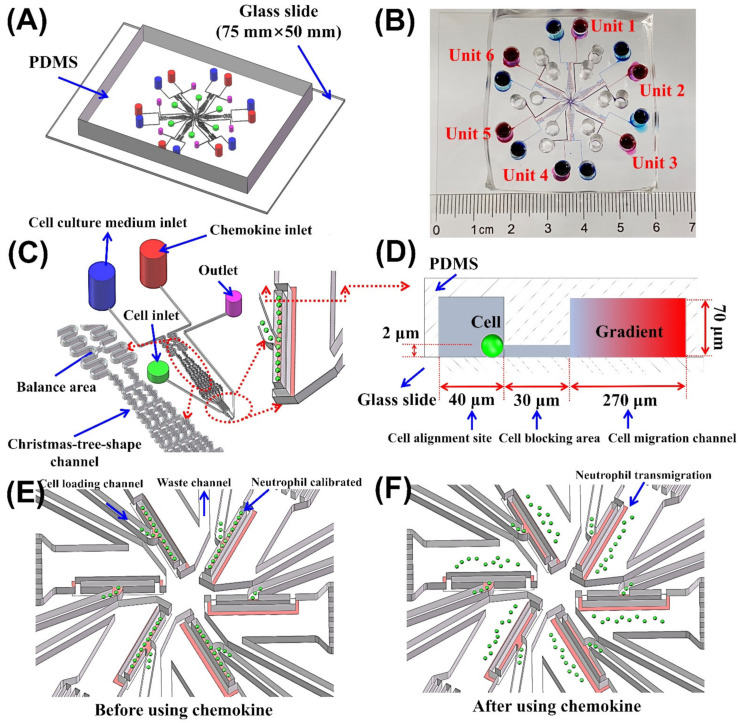
Overall structure diagram of the SC^2^ chip. (**A**) Equiaxial side view of the SC^2^ chip; (**B**) Front view of the SC^2^ chip (photos). (**C**) Enlarged view of an independent test unit. (**D**) Enlarged view of the cell arrangement area. (**E**) Cell distribution in the SC^2^ chip before using chemokine. (**F**) Cell distribution in the SC^2^ chip after using chemokine for 15 min.

**Figure 3 micromachines-13-01820-f003:**
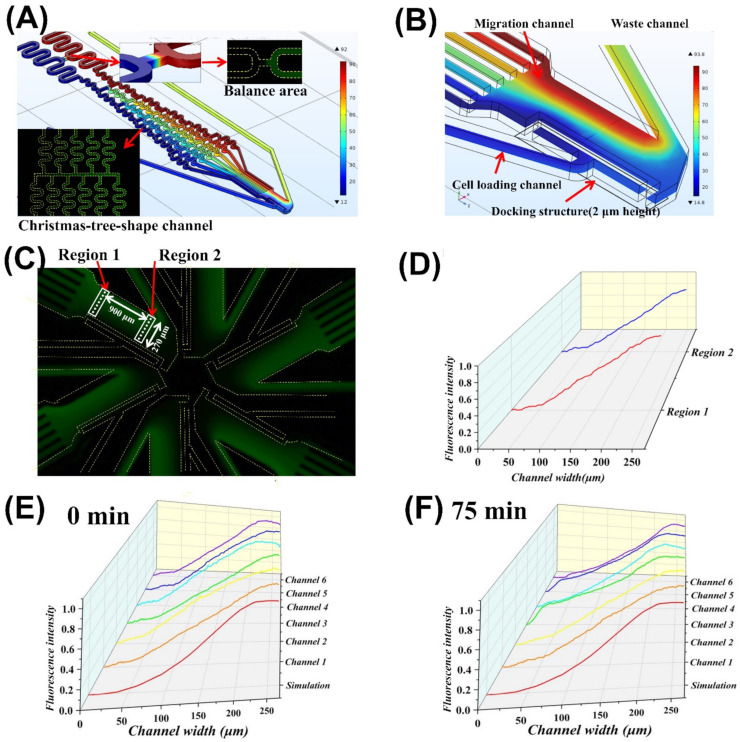
Gradient simulation and test. (**A**) A simulation heat map of an independent unit, including the tested fluorescence images at the balance area and the Christmas-tree-shape channel of the SC^2^ chip; (**B**) Enlarged gradient heat map in the migration channel; (**C**) The fluorescent image of the six migration channels through single imaging using the microscope; (**D**) Comparison of the fluorescence gradient curves at two different positions in one migration channel as labeled in (**C**); (**E**) Comparison of the simulative and experimental fluorescent gradient profiles along the migration channel at 0 min; (**F**) Comparison of the simulative and experimental fluorescent gradient profiles along the migration channel at 75 min.

**Figure 4 micromachines-13-01820-f004:**
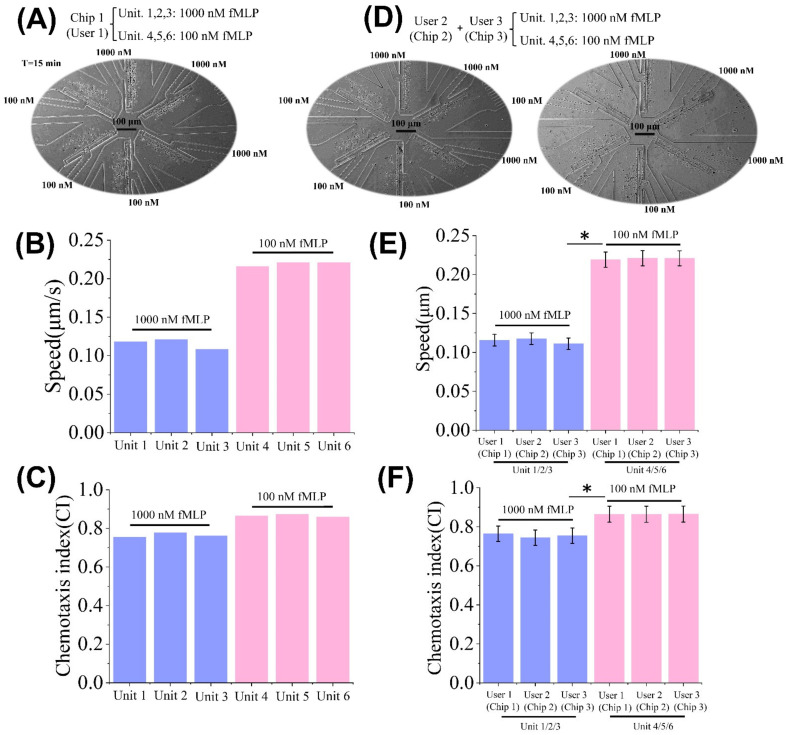
Validation of all-on-chip chemotaxis tests. (**A**) Experiment images in the SC^2^ chip after 15 min. (**B**) The migration speeds of magnetically sorted neutrophils in different units. (**C**) The CI of magnetically sorted neutrophils in different units. (**D**) Experiment setup and images for validating the consistency of different SC^2^ chips and user-friendliness. (**E**) The migration speeds of magnetically sorted neutrophils from different users (chips). (**F**) The CI of magnetically sorted neutrophils from different users (chips). The statistical significance was assessed of the difference between two sets of data using the Mann-Whitney test. * means a significant difference (*p* < 0.05).

**Figure 5 micromachines-13-01820-f005:**
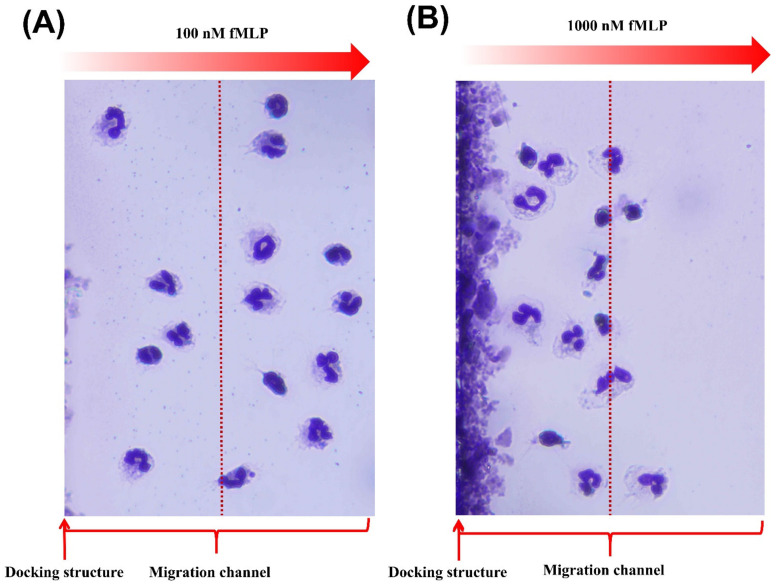
Giemsa images of the neutrophils in the migration channel. (**A**) The cell images at 15 min in 100 nM fMLP gradient. (**B**) The cell images at 15 min in 1000 nM fMLP gradient.

**Figure 6 micromachines-13-01820-f006:**
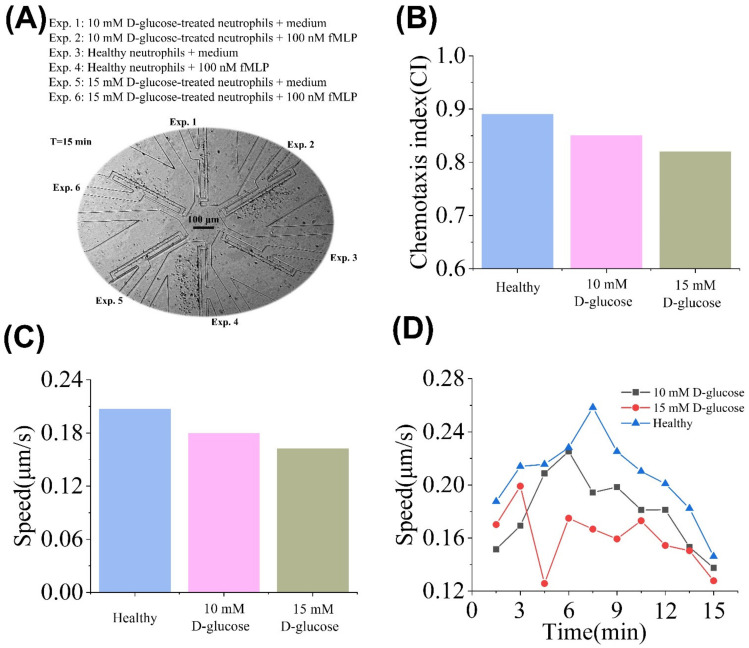
Chemotaxis comparison of the D-glucose-treated and control neutrophils. (**A**) Experiment images in the SC^2^ chip after 15 min. (**B**) Average CI values of healthy neutrophils, 10, 15 mM of D-glucose-treated neutrophils. (**C**) Average V values of healthy neutrophils, 10, 15 mM of D-glucose-treated neutrophils. (**D**) Time-dependent V values of healthy neutrophils, 10, 15 mM of D-glucose-treated neutrophils.

**Figure 7 micromachines-13-01820-f007:**
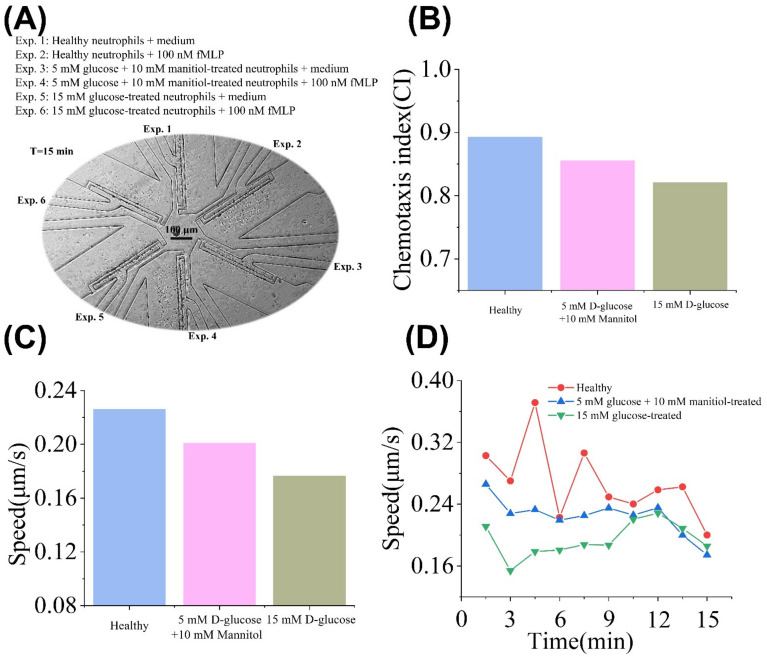
Chemotaxis of the healthy neutrophils, 15 mM D-glucose, and 5 mM D-glucose + 10 mM mannitol pretreated neutrophils. (**A**) Experiment setup and images in the SC^2^ chip after 15 min. (**B**) Average CI values of the healthy neutrophils, D-glucose (5 mM) + mannitol (10 mM), and D-glucose (15 mM) pretreated neutrophils. (**C**) Average V values of the healthy neutrophils, D-glucose (5 mM) + mannitol (10 mM), and D-glucose (15 mM) pretreated neutrophils. (**D**) Time-dependent V values of the healthy neutrophils, 15 mM D-glucose, and 5 mM D-glucose + 10 mM mannitol pretreated neutrophils.

**Figure 8 micromachines-13-01820-f008:**
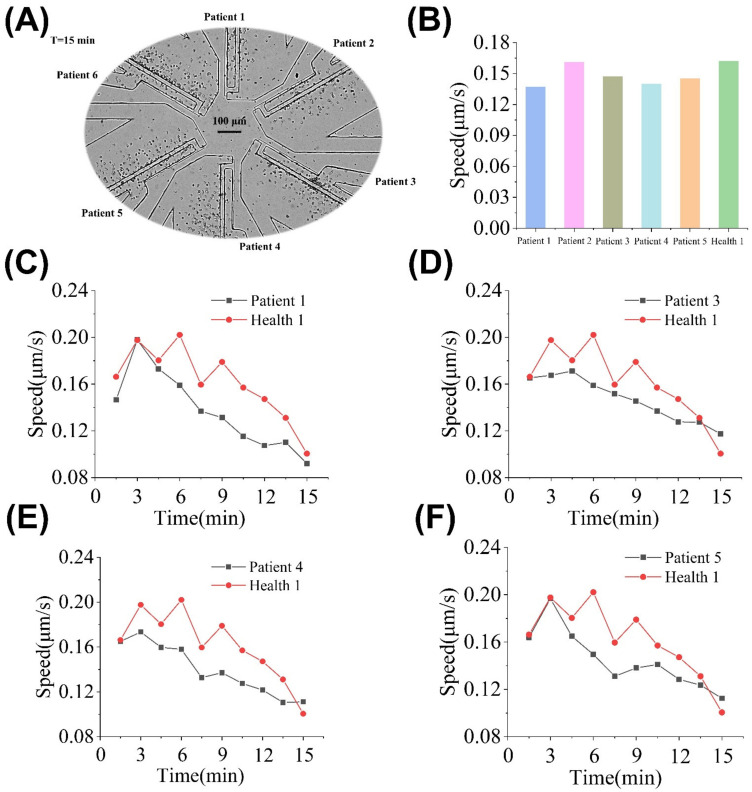
Neutrophil chemotaxis of TD2M patients (n = 5) and healthy person (n = 1). (**A**) Experiment setup and images in the SC^2^ chip after 15 min. (**B**) Average V values of healthy 1 and five TD2M patients. (**C**) Time-dependent V values of neutrophils of healthy 1 and TD2M patient 1. (**D**) Time-dependent V values of neutrophils of healthy 1 and TD2M patient 3. (**E**) Time-dependent V values of neutrophils of healthy 1 and TD2M patient 4. (**F**) Time-dependent V values of neutrophils of healthy 1 and TD2M patient 5.

**Figure 9 micromachines-13-01820-f009:**
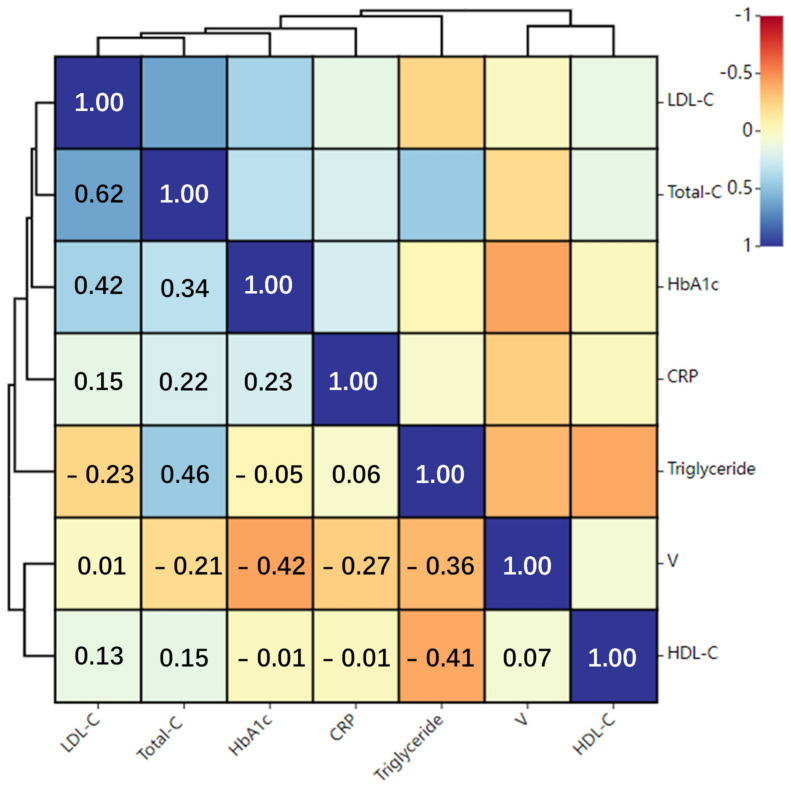
Correlation analysis of cell migration speed with physiological indicators.

**Table 1 micromachines-13-01820-t001:** The serum levels in the “well-control” and “poor-control” TD2M patients.

Good Control	HbA1c (%)	CRP (mg/L)	Total-C (mmol/L)	HDL-C (mmol/L)	LDL-C (mmol/L)	Triglyceride (mmol/L)	V (μm/s)
Patient 1	7.86	0.50	5.87	1.69	3.02	1.73	0.15
Patient 2	7.77	0.50	4.52	0.87	2.42	2.03	0.17
Patient 3	7.50	0.50	1.08	1.00	1.50	1.03	0.14
Patient 4	7.70	0.50	4.17	1.36	2.22	1.07	0.13
Patient 5	6.80	1.48	5.28	1.42	2.84	1.52	0.15
Average	7.53	0.70	4.18	1.27	2.40	1.48	0.15
Poor control	HbA1c (%)	CRP (mg/L)	Total-C (mmol/L)	HDL-C (mmol/L)	LDL-C (mmol/L)	Triglyceride (mmol/L)	V (μm/s)
Patient 6	9.90	2.62	5.41	1.20	2.85	2.00	0.12
Patient 7	8.40	2.24	4.15	1.33	2.22	0.71	0.11
Patient 8	8.30	1.02	6.28	0.97	1.72	8.48	0.10
Patient 9	8.59	0.78	4.00	1.81	1.62	0.83	0.14
Patient 10	10.40	0.50	5.72	1.26	3.51	1.09	0.09
Patient 11	9.10	1.04	5.22	1.26	3.01	1.15	0.18
Patient 12	8.60	0.50	3.82	1.29	1.94	0.68	0.15
Average	9.04	1.24	4.94	1.30	2.41	2.13	0.13

## Data Availability

Data is contained within the article or Appendix A.

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
