# Peer review of "Simplified Cell Magnetic Isolation Assisted SC2 Chip to Realize “Sample in and Chemotaxis Out”: Validated by Healthy and T2DM Patients’ Neutrophils"

_micromachines, 2022, doi:10.3390/mi13111820_

Round 1

Author Response

Pls see the attachment.

Reviewer 2 Report

The article "Simplified cell magnetic isolation assisted SC2 chip to realize sample in and chemotaxis out: validated by healthy and 3 T2DM patients' neutrophils" by Xiao YangItet et al. addresses rather interesting and relevant issues, namely: the deepening of knowledge of microfluidics as a biological model applied to neutrophil chemotaxis in patients with type 2 diabetes mellitus. They have developed a chemotaxis method on blood samples that allows analysis of healthy neutrophils and disease-associated neutrophils. A very well-designed experiment carried from the  benchtop to the clinic.

The microfluidic chip, also known as microfluidic lab-on-a-chip, is a new tool for rapid basic chemical and biological analysis by controlling fluids in networks of microchannels on a chip of a few square centimeters, with which many key laboratory operations, such as sample injection, separation, assay, mixing, reaction, and biochemical detection, can be performed. With the development of life sciences and analytical chemistry, most applications require devices that can be used as disposables in order to eliminate the risks of sample contamination. However, expensive glass, quartz or silicon chips cannot meet this demand. Therefore, it is very important to develop new more advanced methods formass production of microfluidic chips at low cost. Polymeric chips and their fabrication methods have proven to be the solution to this challenge. In fact, tumors studies using biomimetic cell culture platforms that offer unique insights into tumor behavior compared with traditional assays give rise relevant advantages. In general, the use of conventional drug screening approaches to develop cancer therapies effective against the intricate degrees of tumor heterogeneity has proven extremely challenging, resulting in poor translational success rates from the bench to clinical trials. Microfluidic devices, such as the TIME-on-Chip are effective surrogates for simulating disease behavior at different stages. These microscale culture platforms require small sample volumes and provide an in vivo-like microenvironment for conducting complex ex-vivo biomimetic assays with increased assay throughput to, at a minimum, exclude drug candidates that would eventually fail during human clinical trials, thus eliminating the need to circumvent ethical issues associated with controversial animal studies.

The authors well described in details their method in terms of its clinically relevant advantages: "less reagent consumption (10 μL blood + 1 μL magnetic beads + 1 μL lysis buffer); less time (5 min cell isolation + 15 20 min chemotaxis assay); no ultracentrifugation; more convenient; higher efficiency; high throughput." The scientific design of the paper is well described and results are clearly reported. The obtained  results correctly support the hypotheses. 

Regarding neutrophils, the authors describe possible applications for T2MD, but there is also a growing recognition of the diversity of neutrophil function and their plasticity in tumor pathology. Emerging evidence indicates that different neutrophil subpopulations, such as low-density neutrophils or polymorphonuclear-derived suppressor cells that resemble neutrophils, are actively involved in cancer growth and metastasis. Since the plasticity of neutrophils allows them to adapt to different tumor microenvironments and exert different effects on cancer development, delineating the heterogeneity of neutrophils and their interaction within TIME could allow the discovery of new mechanisms of metastasis and help develop appropriate immunotherapies targeting neutrophils and their specific subtypes. However, the cancer-associated biomolecular cues that drive neutrophil function within a primary breast cancer niche are not well defined.

The paper can be accepted for publication on Micromachine. The authors are just invited to discuss in the Conclusion section the possible areas of application of the proposed methods in pathological contexts other than samples from T2DM patients and whether the sample method of chemotaxis can be applied to other circulating cells.

Author Response

Pls see the attachment.
